# Metabolic Consequences of Glucocorticoid Exposure before Birth

**DOI:** 10.3390/nu14112304

**Published:** 2022-05-30

**Authors:** Abigail L. Fowden, Owen R. Vaughan, Andrew J. Murray, Alison J. Forhead

**Affiliations:** 1Department of Physiology, Development and Neuroscience, University of Cambridge, Cambridge CB2 3EG, UK; amj267@cam.ac.uk (A.J.M.); ajf1005@cam.ac.uk (A.J.F.); 2EGA Institute for Women’s Health, University College London, London WC1E 6HX, UK; o.vaughan@ucl.ac.uk; 3Department of Biological and Medical Sciences, Oxford Brookes University, Oxford OX3 0BP, UK

**Keywords:** glucocorticoids, cortisol, placenta, fetus, metabolism, nutrients

## Abstract

Glucocorticoids have an important role in development of the metabolic phenotype in utero. They act as environmental and maturational signals in adapting feto-placental metabolism to maximize the chances of survival both before and at birth. They influence placental nutrient handling and fetal metabolic processes to support fetal growth, fuel storage and energy production with respect to nutrient availability. More specifically, they regulate the transport, utilization and production of a range of nutrients by the feto-placental tissues that enables greater metabolic flexibility in utero while minimizing any further drain on maternal resources during periods of stress. Near term, the natural rise in fetal glucocorticoid concentrations also stimulates key metabolic adaptations that prepare tissues for the new energy demanding functions after birth. Glucocorticoids, therefore, have a central role in the metabolic communication between the mother, placenta and fetus that optimizes offspring metabolic phenotype for survival to reproductive age. This review discusses the effects of maternal and fetal glucocorticoids on the supply and utilization of nutrients by the feto-placental tissues with particular emphasis on studies using quantitative methods to assess metabolism in rodents and sheep in vivo during late pregnancy. It considers the routes of glucocorticoid overexposure in utero, including experimental administration of synthetic glucocorticoids, and the mechanisms by which these hormones control feto-placental metabolism at the molecular, cellular and systems levels. It also briefly examines the consequences of intrauterine glucocorticoid overexposure for postnatal metabolic health and the generational inheritance of metabolic phenotype.

## 1. Introduction

In adults, glucocorticoids (GCs) are stress hormones with a range of metabolic effects which aid survival in environmental conditions that challenge homeostasis [1]. They maintain an energy supply to key tissues when nutrients and oxygen (O_2_) are either scarce or in increased demand such as during pregnancy. In the fetus, GCs have a more diverse range of metabolic functions [2]. They act to modify feto-placental metabolism with respect to resource availability, particularly when availability of nutrients and O_2_ is limited [2]. In addition, towards term, they initiate many of the normal metabolic adaptations that are essential for a successful transition to extrauterine life [3]. The GCs, therefore, have an important role in optimizing metabolic development for survival both before and at birth. This review discusses the role of GCs in programming metabolic phenotype in utero with a particular emphasis on in vivo studies in rodents and sheep. More specifically, it focuses on in vivo studies quantifying the effects of maternal and fetal glucocorticoids on the supply and utilization of nutrients by the feto-placental tissues in experimental animals during late gestation with particular emphasis on rodents and chronically instrumented sheep. Where appropriate, it also makes reference to in vitro human data largely using perfused placenta and placental explants. The review also considers the routes of GC exposure during pregnancy and the mechanisms by which GCs act on feto-placental metabolism at the molecular, cellular and systems levels. Finally, it briefly examines the consequences of the GC-induced changes in fetal-placental metabolism for both fetal development more generally and the longer term programming of postnatal metabolic health and disease.

## 2. Feto-Placental Glucocorticoid Exposure

In most species studied to date, maternal glucocorticoid (GC) concentrations rise during pregnancy and show a diurnal rhythm that relates, in part, to the type and pattern of nutritional intake [4,5,6,7]. Maternal GC concentrations also increase with the number of fetuses and are higher than the corresponding fetal concentrations for much of pregnancy [7]. As GCs are fat soluble, maternal GCs cross the placenta down the concentration gradient and can account for as much as 80% of the GCs circulating in the fetus at mid-gestation [8]. Fetal GC concentrations, therefore, normally vary directly with the maternal values for much of pregnancy [3]. Feto-placental GC exposure also increases when maternal GC concentrations rise in response to psychological and/or physiological stressors, such as undernutrition, isolation and transport (Figure 1).

However, in all animals studied to date, the placenta contains the enzyme, 11β-hydroxysteroid dehydrogenase type 2 (11βHSD2), which metabolizes active GCs to their inactive keto forms and, thereby, limits the extent of feto-placental exposure to the higher maternal GC concentrations [15]. Placental 11βHSD2 activity varies with species, gestational age and environmental conditions, such as undernutrition and hypoxia and is, therefore, a key determinant of feto-placental GC exposure, independent of the maternal GC concentrations [16].

By late gestation, the fetal hypothalamic-pituitary-adrenal (HPA) axis is functional and can respond directly to a variety of stressful stimuli including hypoglycemia and hypoxia induced by conditions such as maternal undernutrition, reduced uterine blood flow, placental insufficiency and occlusion of the umbilical cord [9,17,18,19]. In sheep and other precocial species, responsiveness of the fetal HPA axis to these challenges increases progressively during late gestation in line with developmental maturation of the axis itself [20,21]. Basal GC concentrations also rise naturally in unstressed fetuses towards term due to increased adrenal GC secretion [3,22]. This prepartum GC increment leads to a wide range of maturational changes in fetal tissues in preparation for their new postnatal functions [3]. The magnitude and time course of the prepartum GC increment, and of the enhanced responses of the fetal HPA axis to stressors, varies with species, multiple pregnancy and nutritional conditions earlier in pregnancy [23,24,25,26]. In fetal sheep, basal cortisol concentrations begin to rise about 2–3 weeks before term and then escalate exponentially in last 5 days before birth to values that exceed the maternal concentrations [22]. This prenatal cortisol surge reflects maturational changes at all levels of the fetal HPA axis from the hypothalamic neurons to the adrenal steroidogenic enzymes [19,23,27]. There is also reduced sensitivity of the HPA axis to negative feedback by circulating GCs near term which contributes to the exponential rise in GC concentrations during the immediate prepartum period in fetal sheep [28]. Furthermore, there are species-specific developmental changes in fetal corticosterone binding globulin (CBG) concentrations and in tissue abundance of GC receptors (GR) and the 11βHSD isoforms 1 and 2, all of which affect fetal GC bioavailability at the cellular level in addition to the circulating fetal GC concentration [16,19,26,29]. Elevated endogenous GC concentrations, therefore, have two major roles in the fetus; first, they act as a sign of adversity for much of gestation and, secondly, they signal the proximity to birth as term approaches [2]. In both situations, the natural GC increment maximizes the chances of offspring survival in response to environmental change, in part through actions on feto-placental metabolism.

In pregnant women, fetal GC overexposure can also be caused exogenously by maternal treatment with synthetic GCs for a variety of clinical conditions such as congenital adrenal hyperplasia, rheumatoid arthritis, asthma, and other inflammatory conditions [30]. Synthetic GCs have also been given for immune suppression to women planning pregnancy either with transplanted organs or using in vitro fertilization (IVF) procedures, which leads to excess embryonic GC exposure from the earliest stages of development [31,32]. In addition, later in pregnancy, antenatal treatment with synthetic GCs is given routinely to women threatened with pre-term delivery to improve neonatal viability of their infants, although in 50% of cases delivery does not actually occur until much nearer to full term [30]. Synthetic GCs are 10–20 times more potent than their natural counterparts and interact only with the GRs unlike the endogenous GCs that bind to both the GR and mineralocorticoid receptors [7]. Since synthetic GCs are also poorly inactivated by placental 11βHSD2, the magnitude of fetal GC overexposure is probably greater with synthetic than natural GCs [16]. Consequently, the metabolic effects of administering synthetic GCs during pregnancy are likely to be more pronounced and differ from those seen with overexposure to endogenous GCs caused by adverse environmental conditions.

Early overexposure of fetal tissues to GCs can, therefore, occur by three main routes: (1) increased transplacental transfer of maternal GC in response to either stress-induced increases in maternal concentrations or decreases in placental 11βHSD2 activity, (2) activation of the fetal HPA axis as a result of an insufficient fetal supply of oxygen (O_2_) or nutrients or both during the last third of gestation or (3), in clinical practice, by maternal administration of synthetic GCs for treatment of pre-existing disorders or threatened preterm labor. Studies in experimental animals have shown that each of these routes of GC overexposure affects metabolism of the placental and fetal tissues with consequences for intrauterine growth and development.

## 3. Effects of Glucocorticoids on Feto-Placental Metabolism

The fetal and placental tissues use a wide range of nutrients for their growth, fuel storage and energy production [33,34]. The nutrients are supplied primarily by the mother but their availability in utero is influenced by conditions not only in the mother but also in the placenta and fetus itself. As environmental and maturational signals [2], the maternal and fetal GCs have a central role in regulating the feto-placental supply and utilization of nutrients in optimizing metabolic fitness for offspring survival in the prevailing intrauterine conditions.

### 3.1. Glucose Metabolism

In all species studied to date, glucose is the principal oxidative substrate of the fetus [33,34]. It also provides carbon for tissue accretion and is stored as glycogen in fetal tissues such as the liver, heart, and skeletal muscle. In addition, it is involved in lipogenesis and contributes indirectly to fat accumulation in fetal adipose and other tissues [33]. Glucose is transported across the placenta by facilitated diffusion down its maternal-fetal concentration gradient using a range of glucose transporters (GLUTs) that are localized to specific placental membranes [33,35]. In addition in late gestation, circulating glucose can be derived from the fetal liver by glycogenolysis of stored glycogen and by gluconeogenesis from other metabolites [36,37].

In sheep, increasing feto-placental GC exposure during late gestation reduces weight specific glucose delivery to the fetus, irrespective of whether treatment is with natural or synthetic GCs, or via the maternal or fetal route (Table 1). 

Umbilical glucose uptake per kg sheep fetus decreases by 30% in response to either fetal or maternal cortisol infusion (Figure 2) and declines as fetal cortisol concentrations rise naturally towards term [3,77]. Weight specific rates of glucose delivery to fetal sheep are therefore inversely correlated to the cortisol concentration in the fetal circulation during late gestation, irrespective of whether the concentration rises naturally, by exogenous infusion or in response to sub-optimal intrauterine conditions such as hypoglycemia [18,71,72,77]. Similarly, in mice dams, oral administration of corticosterone for 5 days during late pregnancy reduces maternal-fetal clearance of labelled non-metabolizable glucose per gram placenta by 30–50% in association with a similar reduction in fetal glucose accumulation (Figure 3). In addition, inducing GC overexposure by deleting the 11βHSD2 gene in mice lowers the rate of placental glucose transfer in late gestation [78]. However, corticosterone treatment for 5 days earlier in mouse pregnancy has little effect on placental transport or fetal accumulation of labelled glucose, despite increases in placental glucose transporter (GLUT) abundance [14]. Acute dexamethasone administration to pregnant rats near term also had no effect on placental transport or fetal uptake of glucose, although more chronic treatment altered the abundance of a wide range of placental nutrient transporters close to term (Table 1). Collectively, these finding indicate that the timing and duration of GC exposure during pregnancy are important factors in determining the placental supply of glucose to the fetus. There is also evidence that the effects of GC exposure on placental glucose transport persist after cessation of exogenous treatment of the fetus, although this may be due to rebound activation of the HPA axis and increased endogenous cortisol concentrations [39,72,79].

Glucocorticoids also affect glucose metabolism in utero through their actions on the glucogenic capacity of the liver and other fetal tissues like the kidney [22,80]. Both synthetic and natural GCs stimulate glycogen deposition in the fetal liver and increase the hepatic activity of key gluconeogenic enzymes involved in glycogenolysis and gluconeogenesis in several species including rodents and sheep [51,54,72,81,82]. The hepatic glucogenic capacity of the sheep fetus, therefore, increases in late gestation in line with the normal prepartum increment in fetal cortisol and enables activation of endogenous glucose production close to term and at birth [37,77]. Similarly, fetal glucose production occurs when fetal cortisol levels rise naturally in response to maternal fasting, insulin-induced hypoglycemia and other stressors in late gestation [18,36,77]. When the fasting-induced increment in fetal cortisol is prevented by fetal adrenalectomy, fetal glucogenesis does not occur despite normal catecholamine concentrations [83]. Short term dexamethasone administration to sheep fetuses close to term has also been shown to stimulate hepatic glucose output in association with increased fetal concentrations of gluconeogenic substrates such as lactate and certain amino acids [66,76]. Endogenous glucose production is, therefore, positively correlated to the cortisol concentration in fetal sheep during the prepartum period and in response to fetal hypoglycemia in late gestation [77,84]. However, neither short periods of dexamethasone administration nor acute fasting-induced increments in fetal cortisol induce fetal glucogenesis earlier in gestation when the hepatic glycogen stores and gluconeogenic enzyme activities are still low [77,85]. Fetal glucose production in response to sustained hypoglycemia is only activated earlier in gestation when cortisol concentrations have risen sufficiently over several days and reached a threshold concentration [77,84] Taken together, the studies indicate that GCs first increase the fetal glucogenic capacity and then activate endogenous glucose production per se after a critical period of GC exposure. The progressive rise in fetal cortisol concentrations over several days before term, therefore, allows a coordinated onset of endogenous glucose production with the decline in umbilical glucose uptake and, thereby, maintains glucose delivery to key feto-placental tissues when the fetal demand is maximal in absolute terms. However, the extent to which activation of fetal glucogenesis by rising fetal cortisol levels is the cause or the consequence of the concomitant fall in umbilical glucose uptake remains unclear.

While maternal or fetal cortisol infusion for a 5-day period in late ovine pregnancy have similar effects on umbilical glucose uptake, their actions on uteroplacental glucose metabolism differ (Figure 2). Uteroplacental glucose consumption increases with fetal, but not maternal, cortisol infusion with no change in uterine glucose uptake with either treatment [62,72]. Conversely, maternal, but not fetal, cortisol treatment increased placental clearance of non-metabolizable glucose from the fetal circulation [62,72]. The cortisol-induced fall in umbilical glucose uptake is, therefore, consistent with the increased uteroplacental glucose consumption with fetal infusion but is likely to reflect changes in other aspects of uteroplacental metabolism with maternal administration.

### 3.2. Metabolism of Lactate and Other Carbohydrates

In several ruminant species, the fetus also uses other carbohydrates, such as lactate and fructose, in addition to glucose, for its growth, glycogen synthesis and oxidative metabolism during normal unstressed conditions [33,80]. These alternative substrates are metabolized from glucose in the uteroplacental and fetal tissues and circulate at relatively high concentrations in fetal sheep during late gestation [61,86,87,88,89]. Their production is GC sensitive and dependent on whether increased GC exposure is of maternal or fetal origin (Table 1, Figure 2). Maternal and fetal administration of cortisol in late gestation have different actions on feto-placental lactate metabolism (Table 1). Maternal cortisol infusion for 5 days increased the weight-specific rates of uteroplacental production and umbilical uptake of lactate (Figure 2), in association with higher maternal but not fetal lactate concentrations [62], although concentrations rise in both circulations when maternal cortisol infusion was more prolonged [61]. In contrast, fetal cortisol infusion for 5 days had no effect on the rates of uteroplacental production and umbilical uptake of lactate (Figure 2) or on the lactate concentrations in either circulation [71,72]. However, fetal administration of the more potent synthetic GC, dexamethasone, for 24 h at a similar gestational age increases umbilical lactate uptake and fetal lactate concentrations [66].

Dexamethasone administered directly to fetal sheep has been shown to increase the use of lactate for hepatic glucose production close to term but not earlier in gestation [76,85]. Maternal cortisol infusion also increases fetal hepatic abundance of lactate dehydrogenase which converts lactate to pyruvate for hepatic gluconeogenesis or oxidation [62]. The uteroplacental switch in carbohydrate provision to the fetus away from glucose towards lactate during maternal hypercortisolemia diversifies the fetal carbon supply and may allow the fetus greater control over its carbon utilization during maternal stress near term. However, only about half of the increase in uteroplacental lactate production induced by maternal cortisol infusion can be accounted for by the fall in the umbilical glucose supply (Figure 2). This suggests that ovine uteroplacental tissues may be using a greater proportion of their normal glucose consumption by glycolysis to produce lactate during maternal hypercortisolemia. Alternatively, these tissues may be producing lactate from glucose or fructose derived from the fetal circulation, consistent with the greater fetal-to-maternal glucose clearance and increased abundance of placental glucose transporter 8 (GLUT8) that transports both glucose and fructose [62].

The source of the extra fructose in the fetal circulation during maternal cortisol infusion remains unclear but may be due to increased fructose synthesis by either the placenta or fetus [62,87,88]. Since fructose can be metabolized oxidatively by the fetus [87], its increased concentration may also contribute to greater metabolic flexibility of the fetus during maternal stress. In addition, converting glucose to other carbohydrates during maternal hypercortisolemia prevents any significant rise in fetal glucose and, hence, insulin concentrations that might otherwise accompany the maternal hyperglycemia. This then has the benefit of minimizing any increase in fetal nutrient demands for insulin-stimulated growth at a time of maternal stress and potential nutrient constraint [90]. Indeed, reductions in fetal insulin levels have been observed in response to maternal treatment with synthetic GCs in both sheep and rodents [57,82,91,92]. Collectively, the changes in lactate production and glucose consumption seen in ovine uteroplacental tissues overexposed to GCs suggest that changes in placental metabolism as well as transport contribute to the reduced fetal glucose supply in these circumstances. The findings also indicate that these tissues adopt different metabolic strategies to supporting offspring survival depending on the magnitude and direction of the transplacental cortisol concentration gradient (Figure 2).

### 3.3. Amino Acid Metabolism

Less is known about the effects of GC exposure on uteroplacental and fetal metabolism of amino acids. Amino acids are required for protein synthesis and can also be used oxidatively by the fetus, particularly in ruminants [33,68]. They are transported across the placenta by active transport using a range of amino acid transporters located in specific placental membranes [93]. The ovine placenta also transaminates certain amino acids derived from the maternal and fetal circulations in supplying essential amino acids to the fetus [33,66]. Short term infusion of cortisol for 6 h into fetal sheep reduces the umbilical uptake of α-amino nitrogen [69]. It also increases proteolysis and decreases protein synthesis in fetal sheep measured using labelled leucine turnover [33,68]. Longer 24 h infusions of dexamethasone into fetal sheep increase the gluconeogenic amino acid concentrations and decrease the umbilical supply of alanine [66]. These changes were accompanied by decreased activity of the placental–hepatic shuttle in fetal glutamine-glutamate metabolism and by diversion of hepatic glutamate uptake into glycogen synthesis [66]. In contrast, fetal cortisol infusion for 5 days appears to have little effect on the umbilical uptake or uteroplacental utilization of α-amino nitrogen although little is known about the placental or hepatic metabolism of individual amino acids in these circumstances [72]. There is an increase in the urea concentration of fetal sheep after 10 days of maternal cortisol infusion, indicative of increased amino acid deamination and oxidative use of amino acids [61]. Transcriptome and metabolome analyses of the ovine placenta after 25 days of maternal cortisol infusion also indicate that there are changes in amino acid metabolism that involve glutamate and the degradation and biosynthesis of branched chain amino acids [64]. Similar changes in the metabolism of branched chain amino acids are also seen in ovine cardiac and skeletal muscle before and at birth after prolonged maternal hypercortisolemia [65,94].

In pregnant mice close to term, administration of corticosterone, but not dexamethasone, in the drinking water led to a 50% reduction in the transplacental transport and fetal accumulation of the non-metabolizable amino acid analogue, methyl aminoisobutyric acid (MeAIB) on a weight-specific basis (Figure 3). When the data from the control and treated dams were combined, placental MeAIB clearance correlated inversely with maternal corticosterone across the physiological range of concentrations in late pregnancy [14]. Earlier in pregnancy, drinking corticosterone increased abundance of the placental System A amino acid transporters and placental MeAIB accumulation but had no effect on transplacental transfer or fetal accumulation of MeAIB [14]. Subcutaneous injection of dexamethasone to mice dams also had little effect on transplacental MeAIB transport in late pregnancy but any effect may have been obscured by the rise in corticosterone concentrations in the saline-treated controls as a result of the stress of daily injections [39]. Individual housing and injection of pregnant rodents are known to affect basal GC concentrations with implications for the metabolic effects of any subsequent experimental treatments (Figure 1, [39,95]). In addition, increased transplacental MeAIB clearance was observed close to term after cessation of oral treatment with either corticosterone or dexamethasone earlier in pregnancy [14,39]. In contrast, decreased System A amino acid transport has been observed in vitro in human placenta delivered several days after completing an antenatal course of synthetic GC treatment [96]. These observations emphasis the importance of the timing, route of administration and type of GC in assessing the metabolic effects of GC exposure in utero.

### 3.4. Oxygen Transfer and Utilisation

Direct manipulation of maternal or fetal GC concentrations in sheep for periods of up to 5 days appears to have little effect on the transfer or utilization of O_2_ by either the uteroplacental or fetal tissues on a weight specific basis, irrespective of the type of GC or timing of treatment in pregnancy [62,66,71,72]. With the concomitant fall in umbilical glucose uptake with GC treatment, less glucose is available for oxidative metabolism which leads to a reduced fetal glucose/oxygen quotient and lowers rates of fetal glucose oxidation [66]. The maintained rate of fetal O_2_ consumption in these circumstances must, therefore, depend on use of other oxidative substrates, particularly before the onset of significant fetal glucogenesis. The increased uteroplacental supply of lactate in response to maternal cortisol infusion will provide an alternative carbon source for fetal oxidation, which almost exactly balances the fall in the glucose carbon supply (Figure 2). The rise in amino acid and urea concentrations in response to maternal and fetal GC treatments also suggests that more amino acids are being used for fetal oxidative metabolism in these circumstances [61,67,68]. Although GCs affect fatty acid oxidation in adulthood, there is little evidence for fatty oxidation in utero, at least in the sheep fetus [33,97]. The finding that O_2_ consumption falls in adrenalectomized but not sham-operated sheep fetuses in response to acute maternal fasting near term suggests that an increment in fetal cortisol is needed for feto-placental tissues to maintain an alternative supply of oxidative substrates during hypoglycemia [83]. Collectively, these observations suggest that GCs have an important role in diversifying substrate availability for fetal oxidative metabolism during stressful conditions. However, in unstressed conditions towards term, activation of fetal glucose production as umbilical glucose uptake falls maintains the normal rates of fetal glucose and O_2_ utilization as there are no changes in rate of glucose carbon oxidation, glucose/oxygen quotient or the fraction of fetal O_2_ consumption used to oxidize glucose, despite rising cortisol levels during the prepartum period [77]. Thus, the GC-induced metabolic adaptations that enhance survival in utero appear to differ from those that ensure viability at birth.

### 3.5. Bioenergetics and Mitochondrial Function

The energy demands for growth and development are high in the fetus and increase still further at birth with the onset of new postnatal functions such as breathing, thermogenesis, digestion, glucoregulation and locomotion [80,97]. Production of energy in the form of adenosine triphosphate (ATP) depends primarily on the functional capacity of mitochondria for oxidative phosphorylation (OXPHOS) using glucose and other sources of carbon and reducing equivalents. Adult mitochondria are dynamic organelles that respond to changes in energy demands by biogenesis and alterations in the electron transfer system (ETS) complexes and other proteins regulating ATP synthesis, including those supporting oxidation of specific substrates [98]. Glucocorticoids are known to influence many of these regulatory processes in adult tissues and recent studies have shown that they also have an important role in the normal prepartum maturation of mitochondrial OXPHOS capacity in cardiac and skeletal muscle [74,99,100,101,102].

Acute administration of synthetic GCs to rats and mice in late pregnancy increases mitochondrial density, respiratory function, and abundance of specific ETS complexes in range of fetal tissues including the brain, liver, kidney, and heart [45,100,102]. Similar increases in mitochondrial volume density are seen in Type II pneumocytes of fetal rabbits after maternal betamethasone injection close to term [103]. Maternal dexamethasone administration to pregnant ewes near term also increases the abundance of mitochondrial ETS and uncoupling proteins in perirenal adipose tissue of their fetuses in line with the need for non-shivering thermogenesis at birth [56,73]. Similarly, direct cortisol infusion into fetal sheep in late gestation increases mitochondrial content and respiratory capacity of skeletal muscle in a manner that depends on the specific oxidative substrate and muscle studied [74]. Furthermore, raising fetal corticosterone indirectly in fetal rats in late gestation by ligating the uterine artery also increases expression of several ETS complexes in a range of fetal tissues two days later [104,105]. Conversely, preventing the natural prepartum rise in fetal GC pharmacologically or by fetal adrenalectomy or gene deletion in the adrenal GC biosynthetic pathway abolishes the normal increase in mitochondrial content in several tissues of fetal sheep and rodents close to term [74,82,102].

In contrast with the effect of short term administration, GC exposure over longer periods appears to suppress mitochondrial respiratory capacity in fetal tissues [41,60,64,65]. Maternal cortisol infusion for last 25 days of ovine pregnancy reduces mitochondrial content and abundance of specific ETS complexes and adenine nucleotide transporters in fetal cardiac and skeletal muscle at term [65]. Likewise, long term placental insufficiency accompanied by decreased umbilical glucose uptake leads to reduced mitochondrial respiratory function and lower ETS Complex I abundance in skeletal muscle of fetal sheep in late gestation [106]. Collectively, these studies show that the timing and duration of GC exposure are both important factors in determining the mitochondrial content and OXPHOS capacity of fetal tissues. Shorter exposures closer to term appear to anticipate the increased postnatal energy requirements by increasing mitochondrial OXPHOS capacity, while longer GC exposures from earlier in gestation limit potential energy production in line with the reduced nutrient supply and fetal growth. Overall, these studies also indicate that Complex I appears to be more responsive than the other ETS complexes to changes in GC exposure.

## 4. Mechanisms of Glucocorticoid Action

### 4.1. Molecular and Cellular Effects

At the molecular level, there are several different GR protein isoforms which have tissue specific patterns of expression [107]. These GRs bind GCs and can have rapid non-genomic actions by interactions with cytoplasmic co-factors controlling cellular metabolic pathways. Alternatively, they can have slower actions on the nuclear and mitochondrial genomes by binding to glucocorticoid response elements (GRE) in the gene promotors which may be local or distant to the target gene [108]. Once the GR is bound to the GRE, it activates the transcription machinery and chromatin modulators to initiate gene transcription. The genomic and non-genomic actions of the GCs can differ between tissues and with development and the stage of the cell cycle due to changes in the expression and interaction of the specific GR isoforms and their post-translational modifications [109]. Glucocorticoid-induced epigenetic modifications include DNA methylation, histone modification and changes in the abundance of non-coding and micro RNAs [110]. Changes in both global and specific gene methylation have been observed in the liver and other fetal tissues after maternal treatment of pregnant rats and guinea pigs with synthetic GCs [108,111]. These studies show that GCs can influence DNA methylation through a range of different mechanisms from alterations in chromatin structure to specific changes in methylation of CpG islands and CpG dinucleotides of the genes.

At the cellular level, GCs alter expression of a wide range of growth factors, enzymes, hormone receptors, ion channels, transporters and proteins of signaling pathways with metabolic effects, often in a tissue-specific manner [2,112,113]. These changes not only alter cell metabolism and bioenergetics in relation to nutrient availability but also prepare tissues for their new metabolic functions and energy requirements after birth [2,3,114]. For instance, GCs alter expression of glucose and amino acid transporters in the hemochorial rodent and human placenta (Table 1, [35]). In rodents, these effects are more pronounced with exposure to natural than synthetic GCs and are dependent on the timing and duration of exposure during pregnancy and on the length of time since cessation of overexposure [14,38,39,59,79]. The actions of GCs on placental nutrient transporters are also sex-linked in rats with increased and decreased GLUT abundance in female and male placentas, respectively, in response to long term dexamethasone administration [49]. In contrast, in the more multilayered epitheliochorial placenta of sheep, there is less evidence for changes in expression of nutrient transporters in response to either maternal of fetal cortisol infusion, despite the alterations in the umbilical supply of nutrients [62,71,72]

The actions of GCs on cellular processes may be mediated, in part, via the mechanistic target of rapamycin (mTOR) pathway known to be responsive to a complex set of metabolic cues including growth factors, hormones and cellular energy availability [115]. This pathway has been shown to stimulate amino acid transport in the human placenta, although less is known about its role in regulating transplacental glucose transport [116]. In pregnant mice fed ad libitum, the decrease in transplacental MeAIB clearance seen in response to corticosterone administration is associated with upregulation of the mTOR inhibitor, DNA damage responses 1 (REDD1) and with reductions in the downstream readouts of mTOR activity [59]. Cortisol also upregulates REDD1 at the mRNA level in human trophoblast cells in vitro [117]. Therefore, maternal glucocorticoids may suppress placental amino acid transport in vivo by reducing amino acid transporter abundance and/or translocation to the trophoblast plasma membrane, via mTOR. Similarly, GC treatment alters expression of proteins in the insulin signaling and mTOR pathways of fetal skeletal muscle in sheep and baboons with divergence between the upstream insulin-sensitive elements and the downstream components relating gene transcription in fetal sheep [53,118]. There are also changes in the insulin responsive Akt pathway in the placenta in the rat placenta following maternal dexamethasone administration [119].

The cellular mechanisms by which GCs inhibit placental glucose transport are less clear as there is little evidence for a consistent reduction of GLUT mRNA or protein in response to GC administration (Table 1). There is also no significant change in the glucose concentration gradient between the ovine maternal and fetal circulations in response to either maternal or fetal cortisol administration, despite maternal and fetal hyperglycemia respectively [62,71,72]. These observations suggest that that there may be changes in the membrane localization or trafficking of specific GLUTs in the placenta in response to GCs exposure. Alternatively, the GC-induced changes in placental metabolism may alter the partitioning or concentration gradients of glucose across the various membranes and compartments forming the placental barrier.

### 4.2. Other Indirect Systemic Effects

While studies of the knock down or deletion of the GR in mutant mice have shown that GCs can act directly in fetal tissues [91,120,121], their molecular and cellular actions could also be secondary to other GC-induced changes in the maternal, placental or fetal environments. Administration of GCs during pregnancy has been shown to alter maternal nutrition, placental morphology, and the concentrations of other metabolic hormones, all of which could mediate, at least in part, the metabolic actions of the GCs [14,122,123,124]. Even the apparent genomic effects of GC exposure in utero may be indirect as maternal treatment of pregnant rats with synthetic GCs has been shown to alter placental transfer of methyl donors essential for DNA methylation [125].

#### 4.2.1. Maternal Nutritional State

Dexamethasone treatment of ewes and rats near term reduces their energy intake [56,126]. In contrast, in both pregnant sheep and mice, administration of the natural GC before term increases food intake and alters glucose-insulin dynamics causing maternal hyperglycemia and hyperinsulinemia with implications for placental nutrient transfer [14,122]. When mice were treated orally with corticosterone and pair-fed back to the control food intake, placental clearance of labelled glucose (but not MeAIB) was restored to normal values (Figure 3). These observations suggest that the actions of corticosterone on placental transport depend, in part, on the maternal nutritional state in a nutrient specific manner. These effects may be mediated, in part, through the mTOR pathway by interactions between placental REDD1 expression and the GC-induced maternal hyperinsulinemia [59]. Nutritional state may also contribute to the alterations in placental nutrient metabolism and transport during maternal hypercortisolemia through changes in GC bioavailability or trafficking of existing transporters to the placental membranes [25,78,79,81,126].

#### 4.2.2. Placental Development

Maternal administration of both synthetic and natural GCs has been shown to restrict placental growth in a range of species with implications for the total placental surface area important in the passive and transporter-mediated diffusion of nutrients and O_2_ [35,93]. Maternal administration of GCs during pregnancy also impairs vascularity of the mouse placenta in a manner that depends on the type and timing of GC exposure [13,14,59,60,89]. In pregnant sheep, maternal cortisol infusion reduces uterine and umbilical blood flows in association with reduced placental weight and an increased proportion of the more everted placentome types [61,122]. In turn, these GC-induced changes in placental morphology may compromise the flow limited mechanisms of transplacental nutrient transport. Indeed, the differing effects of maternal administration of corticosterone and dexamethasone on MeAIB clearance in the mouse placenta may reflect, in part, the differences in placental vascularity induced by the two treatments, despite the similarity in placental growth restriction [14,39]. Similarly, in ewes infused with cortisol, the fall in umbilical glucose uptake may relate to the increased eversion of the placentomes [61] as a higher proportion of everted placentomes is associated with a greater fall in umbilical glucose uptake per kg placenta in response to elevating fetal cortisol levels [124]. Although GC administration directly to fetal sheep in late gestation appears to have less pronounced effects on placental growth per se, there are still specific changes in placental morphology including reduced placentome eversion and lower numbers of binucleate cells that produce ovine placental lactogen (oPL) [124,127].

#### 4.2.3. Other Metabolic Hormones

Some of the metabolic actions of the GCs are mediated, in part, via other hormones and growth factors [123]. Glucocorticoids are known to alter functioning of several endocrine systems in fetal sheep including the somatotrophic axis, adreno-medullary catecholamine secretion, the orexigenic-anorexigenic regulatory system and thyroid hormone bioavailability as well as influencing activity of the HPA axis itself [128,129,130,131,132,133,134,135]. Consequently, preterm and natural prepartum increases in fetal cortisol concentrations cause increases in the circulating concentrations of triiodothyronine (T_3_), adrenaline, leptin and insulin-like growth factor (IGF)-I in fetal sheep. In turn, these cortisol-dependent endocrine changes contribute to maturation of some of the metabolic processes essential for neonatal survival such as thermoregulation, endogenous glucose production and appetite regulation [114,123]. For example, the prepartum increases in hepatic glycogen content and gluconeogenic enzyme activities, are attenuated by abolishing the cortisol-induced rise in plasma T_3_ concentrations by fetal thyroidectomy [136,137]. Similarly, there is no prepartum increase in mitochondrial density and OXPHOS capacity in skeletal muscle of thyroidectomized sheep fetuses [138]. In addition, GC administration to either the ewe or fetus affects production and secretion of several placental hormones with metabolic effects in the mother, such progesterone and oPL, which, in turn, affect the feto-placental supply of metabolic substrates indirectly [122,127].

## 5. Effects of the Glucocortocid-Induced Metabolic Changes on Fetal Development

The GC-induced changes in feto-placental metabolism switch tissues from accretion to differentiation, irrespective of whether the GC increment occurs naturally towards term or by exogenous administration earlier in gestation [2,3,61,139]. While this switch aids survival, it reduces the fetal growth rate overall and can lead to intrauterine growth restriction, if GC exposure is prolonged before term. Consequently, preterm administration of synthetic and natural GCs is associated with reduced birth weight or body weight at term in a wide range of species including human infants [140,141]. Indeed, in fetal sheep, reductions in the daily crown rump and girth increment are seen with periods of GC administration as short as 5 days [61,139]. However, while growth of most fetal tissues is impaired by exogenous GC administration, there is evidence for brain sparing in the short term, which may be due to the preferential distribution of glucose and oxygen to the brain at the expense of more peripheral tissues following GC-induced changes in regional blood flow [142].

The GC-induced onset of tissue differentiation during the prepartum period is essential for acquiring the tighter glucoregulation and greater central control of metabolism needed after birth as the fetus transitions from the continuous placental supply of nutrients in utero to the more intermittent pattern of enteral nutrition after birth. In the interim period between the loss of the placenta and the establishment of nutritive sucking, the neonate is entirely dependent for survival on its fuel reserves and functional mechanisms for utilizing them for energy production [97]. For instance, the cortisol-induced increase in hepatic glycogen deposition, gluconeogenic enzyme activities and β-adrenoreceptor density enhances responsiveness of the glucogenic pathways to the elevated catecholamine concentrations seen at birth [37]. Likewise, the GC-induced increases in expression of the mitochondrial uncoupling proteins, uncoupling protein 1 (UCP1), and the voltage dependent anion channel upregulate the thermogenic capacity of brown adipose tissue deposits for neonatal heat production [56,73]. Furthermore, in fetal muscles exposed to GCs during late gestation, the increase in insulin-responsive glucose transporter, glucose transporter type 4 (GLUT4), and the relative reduction in the non-insulin-responsive glucose transporter type 1 (GLUT1) proteins underpin the switch of insulin from acting as a growth regulatory hormone in utero to its major glucoregulatory role after birth [53,55,75]. Moreover, GCs affect the development and stress responsiveness of the HPA axis per se as well as regulating their own cellular bioavailability by effects on GR abundance and activity of the 11βHSD1 and 2 isoforms in several fetal tissues [57,129,133,143,144,145]. Indeed, in sheep, the GC-induced decrease in placental 11βHSD2 increases placental cortisol exposure and, thereby, heightens the GC-dependent prepartum changes in placental steroidogenesis and prostaglandin synthesis that control myometrial contractions and the onset of labor in ruminants [122,133,146]. The GCs, therefore, act in utero not only to regulate the metabolic profile of individual tissues but also to coordinate their metabolic activities functionally to ensure a metabolic phenotype fit for future homeostatic challenges.

## 6. Postnatal Metabolic Consequences of Prenatal Glucorticoid Overexposure

While the normal prepartum rise in fetal cortisol is essential for the metabolic adaptations to extrauterine life, early exposure to GCs can have adverse metabolic consequences later in juvenile and adult life, despite aiding neonatal survival should delivery occur prematurely [146,147,148,149]. By inducing a premature switch from tissue accretion to differentiation, early overexposure to GCs can reduce cell number, alter cell composition, and activate functional changes in tissues that are inappropriate for the stage of development with implications for postnatal metabolism, particularly as the functional reserve declines with increasing age. Metabolic dysfunction has been seen in both juvenile and adult animals after experimental GC overexposure in utero in a wide range of species. These studies have shown postnatal abnormalities in appetite, adiposity, glucose tolerance, insulin sensitivity, mitochondrial function and in hepatic gluconeogenic capacity [81,104,135,150,151,152,153,154,155,156,157,158,159,160,161,162,163]. For instance, the increased activity of key hepatic gluconeogenic enzymes induced in the fetus by excess GCs persists into adulthood in rats and sheep [54,156]. The metabolic dysregulation tends to become more pronounced with aging and may lead to adult metabolic diseases such as Type 2 diabetes and Metabolic Syndrome that shorten lifespan in human populations. The adult metabolic dysfunction programmed by GCs in utero can also be sex-linked and may only emerge after a postnatal triggering event, or second hit, such as puberty, pregnancy, or a change in diet [164,165,166].

Prenatal GC overexposure also leads to postnatal HPA dysfunction with implications for metabolic control long after birth. Changes in basal GC concentrations and HPA responsiveness to adrenocorticotropic hormone (ACTH), corticotrophin-releasing hormone/arginine vasopressin and to psychological stresses have been observed in the offspring of mice, rats, guinea pigs, sheep, and non-human primates after maternal GC treatment [27,146,157,158,161,162,167,168,169]. The altered responsiveness is associated with changes at all levels of the HPA axis including the number and activity of hypothalamic neurons, pituitary ACTH production and secretion, and adrenal size and steroidogenic capacity. Increases in cortisol secretion in response to stressors are also seen in children born at term after their mother received antenatal GC treatment [170,171,172,173]. Similar changes in postnatal HPA responsiveness are also seen with age in sheep after either maternal or direct fetal treatment with synthetic GCs [156,157]. Collectively, these studies show that HPA responsiveness is enhanced in juveniles and depressed in adults after maternal GC treatment, although responses depend, in part, on the sex of the offspring and the maternal GC treatment regime. Moreover, prenatal GC exposure has tissue-specific effects on GR, 11βHSD isoforms and CBG abundance after birth [15,140,158]. The metabolic dysregulation seen in later life after prenatal GC overexposure is, therefore, likely to reflect not only the changes in the tissue structure and function induced in utero but also the impact of altered endogenous secretion and bioavailability of GCs in extrauterine life.

The metabolic effects of prenatal GC overexposure are not confined to the immediate offspring but are also observed in subsequent generations. Increased adiposity, dyslipidemia, hepatic phosphoenolpyruvate carboxykinase (PEPCK) activity and impaired insulin-glucose dynamics have been observed in both the F1 and F2 generations after F0 treatment with synthetic GCs in pregnant rats, guinea pigs, sheep, and baboons [79,81,111,147,153,161,164,166,170,174,175]. The metabolic dysregulation can be transmitted by either the paternal or maternal line and is seen in both the male and female offspring [111,176,177]. In part, these intergenerational effects may reflect changes in placental phenotype as MeAIB clearance, amino acid transporter abundance and expression of imprinted, growth regulatory genes are altered differentially in the F2 rodent placenta depending on whether the F1 mother or father was overexposed to GCs in utero [79,111]. In the F3 generation not exposed directly to the original F0 glucocorticoid treatment, the abnormalities in glucose tolerance and hepatic PEPCK activity seen in previous generations are resolved in rats [111]. In contrast, the dyslipidemia induced in F1 adult female baboons by maternal stress persists to the F3 generation via the female lineage [178]. Similarly, in juvenile guinea pigs, the increased HPA response to stress induced by maternal treatment with synthetic GCs is maintained to the F3 generation via both the maternal and paternal lineages, although the cortisol increment diminishes in each successive generation [179]. These effects were associated with generation-specific methylation signatures and differential gene expression in the hippocampus and pre-frontal cortex of all three generations but with no generational overlap between the affected genes [180,181]. Consequently, the extent to which the transgenerational effects of GC overexposure on metabolism are due to inherited epigenetic changes or de novo programming by the parental metabolic dysfunction in each successive generation still remains unclear [148,182,183].

## 7. Conclusions

Glucocorticoids have a central role in the bidirectional metabolic communication between the mother and fetus in experimental animals in vivo. They adapt placental nutrient handling and the fetal metabolic phenotype to balance resource availability with the genetically determined drive for fetal growth [2]. When maternal GC concentrations rise before term in sheep and rodents, the metabolic adaptations support the needs of the existing fetal mass, while limiting further resource demands on the stressed mother by slowing feto-placental growth. When the fetus but not the mother is stressed, the specific rise in fetal glucocorticoid concentrations increases the capacity of the sheep fetus to use its fuel reserves to produce glucose endogenously and to provide alternative substrates for oxidative metabolism. A greater amount of uterine glucose uptake can then be used to maintain the metabolic and endocrine functions of the placenta essential to fetal survival. The decrease in tissue accretion induced by the rise in fetal GC concentrations also reduces the overall demand for nutrients and O_2_ when the availability of these substances may be limited. With both maternal and fetal stress, the GC increments, therefore, induce a greater metabolic flexibility in supplying the existing feto-placental tissues with energy in experimental animals.

In unstressed conditions near term, fetal glucocorticoids trigger maturational changes in fetal metabolic processes in preparation for the new postnatal functions and more integrated metabolic control required to respond to postnatal homeostatic challenges [3]. Maturation of these metabolic processes can also be activated by stress-induced GC overexposure before term, which improves the chances of neonatal survival should delivery occur prematurely. Consequently, in both stressed and unstressed conditions, GCs act to optimize metabolic development for offspring survival to reproductive age. While the ensuing metabolic phenotype is honed to improve offspring fitness, it may become maladaptive postnatally and lead to adult metabolic dysfunction, particularly if the postnatal environment differs from that signaled by GC overexposure before birth. However, the molecular mechanisms by which GC overexposure affects the feto-placental metabolic phenotype and its inheritance to the next generation and beyond remain largely unknown; nor is it clear to what extent this metabolic programming is sex-linked before birth or modifiable by prenatal interventions. Further studies are, therefore, need to determine the full range of genetic, epigenetic, endocrine and other systemic factors that contribute to the metabolic programming induced by GC overexposure in utero, particularly with regard to the long term consequences for human infants whose mother receive antenatal treatment with synthetic glucocorticoids [182,183,184,185]. Whatever the mechanisms involved, a fine balance exists between the beneficial effects of intrauterine GCs for survival in early life and the possibility of more detrimental metabolic outcomes in adulthood, which has important implications for the clinical use of synthetic glucocorticoids during human pregnancy [184,185].

## Figures and Tables

**Figure 1 nutrients-14-02304-f001:**
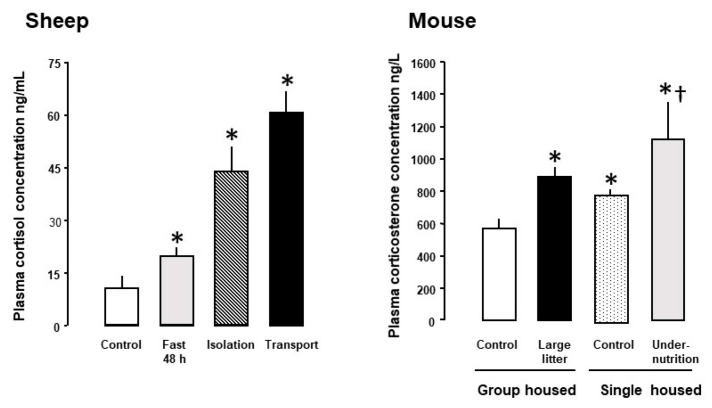
Mean (± standard error (SE)) maternal glucocorticoid concentrations in response to environmental challenges with to normal husbandry conditions in sheep and mice. For sheep, normal conditions are ad libitum feeding in groups in a barn or pen compared to fasting for 48 h in the sight and sound of other sheep, isolated without sight or sound of conspecifics, or transported for 2 h with other sheep. For mice, the controls were ad libitum fed and housed either as a group or singly within sound of other mice or singly housed within sound of other mice and undernourished by feeding at 80% of the control intake. The large litter group housed dams had ≥ 10 pups compared to the other groups with 6–7 pups. Sheep: * Significantly different from their respective controls *p* < 0.05. Mouse: * Significantly different from the group housed control, *p* < 0.05; † Significantly different from single housed control, *p* < 0.05. Data from references [9,10,11,12,13,14] and unpublished observations.

**Figure 2 nutrients-14-02304-f002:**
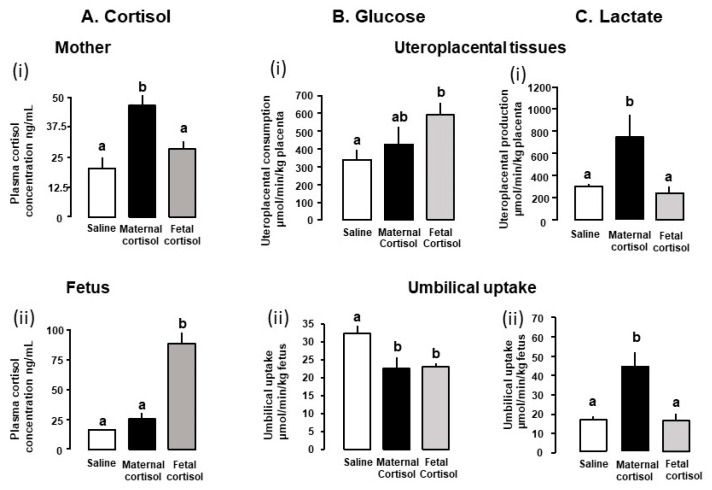
Mean (±SE) values of (**A**) cortisol concentrations in the mother (**A**i) and fetus (**A**ii), (**B**) glucose consumption by the uteroplacental tissues (**B**i) and umbilical uptake by the fetus (**B**ii) and (**C**) lactate production by the utero placental tissues (**C**i) and umbilical uptake by the fetus (**C**ii) after infusion of saline (open columns) or cortisol into the mother (black columns) or fetus (grey columns) from 125 to 130 days of ovine pregnancy (term = approximately 145 days). Within each panel, columns with different letters are significantly different from each other (one-way ANOVA, *p* < 0.05). Data from references [62,71,72].

**Figure 3 nutrients-14-02304-f003:**
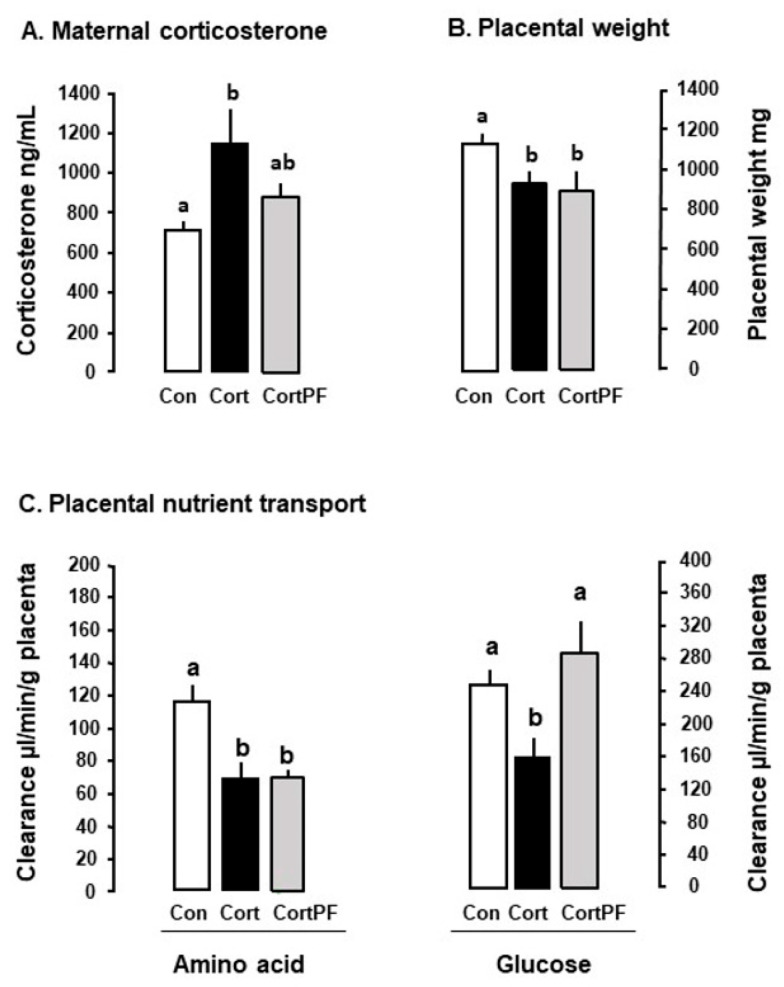
Mean (±SE) values of (**A**) maternal corticosterone concentrations, (**B**) placental weight and (**C**) placental nutrient transport measured as clearance of labelled non-metabolizable glucose or an amino acid analogue measured at day 19 of mouse pregnancy (term = 20 days) in control mice fed ad libitum (open columns, Con), mice treated with corticosterone in their drinking water and fed ad libitum (black columns, Cort) and mice treated with corticosterone in their drinking water and pair fed back to the intake of the control mice (grey columns, CortPF). Within each panel, columns with different letters are significantly different from each other (one-way ANOVA, *p* < 0.05). Data from references [14,59].

**Table 1 nutrients-14-02304-t001:** Effects of glucocorticoid treatments during the second half of pregnancy on feto-placental metabolism of sheep and rodents in vivo.

Treatment	Species	Period of Treatment	Gestational Age at Study	Metabolic Effects	Reference
Maternal Treatments					
Synthetic Glucocorticoids					
Dexamethasone	Mouse	14 and 15 day sc	19 day	↓ Placental MeAIB transport No Δ placental Slc38 a1, Slc38 a2 or Slc38 a4	[38]
		11–16 days po	19 day	↑ Placental MeAIB clearance ↓ Slc38 a2	[39]
		11–16 days sc	19 day	No Δ placental MeAIB clearance ↓ Slc38 a2	
		14–19 days po	19 day	No Δ placental MeAIB clearance Maternal hyperglycemia	
		14–19 days sc	19 day	No Δ placental MeAIB clearance Maternal hyperglycemia	
		16 day sc	17 day	↓ Placental glycogen ↑ Fetal hepatic glycogen content	[40]
		17 day ip	18 day	Fetal heart: ↓ fatty acid translocase ↓ fatty acid oxidation genes ↓ PGC1α (regulator of mitochondrial biogenesis) No Δ mitochondrial morphology	[41]
	Spiny mouse	21 day sc	23 day 37 day	↓ placental GLUT1 expression Sex-linked ↓ placental glycogen Females: ↑ GLUT1 Males: ↓ GLUT1	[42,43]
	Rat	13–21 days po	22 day	Fetal skeletal muscle: ↓ protein content ↓ protein synthetic rate, Fetal Liver: ↑ tyrosine aminotransferase expression	[44]
		14 and 15 day im	16 day	Fetal Brain: ↑ ADP stimulated mitochondrial respiration	[45]
		16, 19 and 21 day sc	22 day	↑ Fetal hepatic and cardiac glycogen	[46]
		2 × daily 18–19 days sc	21 day	Fetal heart: ↑ ATP synthase activity, ↑ ATP content, ↑ glycolysis enzymes, ↑ pyruvate content	[47]
		2 × daily 18–20 days sc	20 day	No Δ placental transport No Δ fetal uptake of glucose	[48]
		9–20 days sc	20 day	Males: ↓ cholesterol, amino acid and TG transporters Females: ↑ cholesterol, TG, amino acid and glucose transporters	[49]
		9–20 days sc	20 day	↓ fetal insulin concentration, ↓ pancreatic β cells ↓ β cell sensitivity to glucose	[50]
		15–20 days sc	20 day	↑ Fetal hepatic glycogen, G6 Pase and PEPCK	[51]
		15–21 days sc Infusion	21 day	↑ Placental GLUT1 and GLUT3 expression	[52]
	Sheep	125 and 126 days im	127 day	Fetal skeletal muscle ↑ GLUT4. Δ No muscle glycogen content	[53]
		125 and 126 days im	127 day	↑ Fetal and maternal hyperglycemia ↑ hepatic glycogen and G6 Pase, No Δ hepatic PEPCK	[54]
		106–107 days im 4 doses	107 day	Maternal and fetal hyperinsulinemia, Fetal hyperglycemia. ↑ Fetal muscle specific GLUT1 and GLUT4	[55]
		138 day im	140 day	Fetal perirenal adipose tissue: ↑ UCP1 and 2 mRNA, VDAC and cytochrome c proteins	[56]
Betamethasone	Sheep	104, 111 and 118 days im	125 day	Decreased fetal insulin concentration with fetal normoglycemia	[57]
		121 day im	122 day	Fetal heart: ↓ PGC1α mRNA (regulator of mitochondrial biogenesis)	[41]
Triamcinolone-acetonide	Rat	16 day ip	21 day	Initial maternal hyperglycemia then hypoglycemia ↓ placental GLUT1 and GLUT3 mRNA and protein	[58]
**Natural Glucocorticoids**					
Corticosterone	Mouse	11–16 days po	16 day	↑ Placental MeAIB accumulation, ↑ Slc38 a1 and Slc38 a2 mRNA ↑ GLUT1 and GLUT3 mRNA	[14]
		11–16 days po	19 day	↑ Placental MeAIB clearance, ↑ Slc38 a1	
		14–19 days po	19 day	↓ Placental MeAIB clearance	
				Maternal hyperinsulinemia ↓ Placental glucose clearance. No Δ GLUT transporters	[59]
		13–15 days minipump	15 day	↓ Placental mitochondrial DNA and Complex III sex-linked	[60]
Cortisol	Sheep	115–130 days iv infusion	130 day	↑ Maternal and fetal lactate concentrations ↑ fetal urea concentrations No Δ α-amino acid concentrations	[61]
		125–130 days iv Infusion	130 day	Maternal hyperglycemia Maternal hyperinsulinemia ↑ Maternal lactate concentration ↓ umbilical glucose uptake ↑ uteroplacental lactate production ↑ increased feto-placental glucose clearance ↑ Placental GLUT8 mRNA ↑ Fetal hepatic lactate dehydrogenase activity No Δ uteroplacental or fetal oxygen consumption	[62]
		115–140 days iv Infusion	140 day	Fetal cardiac muscle: ↓ mitochondrial DNA	[63]
				Altered placental metabolism of glutamate, branched chain amino acids and glycerophospholipids	[64]
				Fetal muscle specific ↓ mitochondrial DNA and metabolism, ↓ cytochrome c ↓ GLUT4 protein, ↓ insulin sensitivity	[65]
**Fetal treatments**					
**Synthetic Glucocorticoids**					
Dexamethasone	Sheep	126 day iv infusion	127 day	↑ fetal gluconeogenic amino acid levels ↓ umbilical alanine uptake ↑ umbilical lactate uptake No Δ umbilical glucose uptake No Δ umbilical oxygen uptake ↓ Placental glutamate uptake from fetal circulation Fetal liver: ↓ Glutamine and gluconeogenic amino acid uptake, ↓ Glutamate output	[66]
		130 day iv infusion	131 day	Fetal and maternal hyperglycemia ↑ fetal gluconeogenic amino acid levels ↓ Placental glutamate uptake from fetal circulation ↓ Umbilical glucose uptake ↓ Glucose/oxygen quotient	[67]
**Natural Glucocorticoids**					
Cortisol	Sheep	121 day iv	121 day +6 h	↑ Fetal proteolysis, ↓ protein accretion, ↑ Fetal leucine oxidation No Δ umbilical glucose or lactate uptake ↓ Umbilical α-amino nitrogen uptake	[68,69]
		122–125 days iv infusion	125 day	↑ Fetal hepatic glycogen content	[22]
		125–130 days iv infusion	130 day	Fetal hepatic G6 Pase, PEPCK, FDP and aminotransferases	[70]
				↓ Umbilical glucose uptake No Δ uteroplacental or fetal oxygen consumption No Δ uteroplacental production of lactate ↓ Umbilical uptake of lactate	[71]
				↓ Umbilical glucose uptake ↑ Uteroplacental glucose utilization No Δ umbilical α-amino nitrogen uptake ↑ Hepatic glycogen deposition ↑ Hepatic mitochondrial pyruvate carboxylase mRNA	[72]
				↑ UCP1 and UCP2 mRNA in fetal adipose tissue	[73]
				Selective fetal skeletal muscles: ↑ mitochondrial density and OXPHOS capacity ↑ Complex I protein ↑ Adenine nucleotide transporter	[74]
		128–130 days iv infusion	130 day	Fetal cardiac muscle: ↓ GLUT1	[75]
		138 d iv infusion	138 day +6 h	↑ Hepatic gluconeogenesis from lactate	[76]

↓ = Decrease compared to control, ↑ = Increase compared to control, Δ = change, GLUT1, GLUT3, GLUT4, GLUT8 = Glucose transporters, OXPHOS = Oxidative Phosphorylation, UCP1, UCP2 = Uncoupling proteins, G6 Pase = Glucose-6-phosphatase, PEPCK = Phosphoenolpyruvate carboxykinase, FDP = Fructose diphosphatase, MeAIB = Methyl Amino-isobutyrate, Slc38 a1, Slc38 a2, Slc38 a4 = System A aminotransferase gene isoforms, TG = Triglycerides, VDAC = Voltage dependent anion channel, PGC1α = receptor gamma coactivator 1-alpha, ADP = adenosine 5′-diphosphate, ATP = Adenosine triphosphate, sc = subcutaneous, po = oral, ip = intraperitoneal, im = intramuscular, iv = intravenous.

## Data Availability

Not applicable.

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
