# Peer review of "Metabolic Consequences of Glucocorticoid Exposure before Birth"

_nutrients, 2022, doi:10.3390/nu14112304_

Round 1

Reviewer 1 Report

This is a narrative review on the effects of maternal and fetal glucocorticoids on the supply and utilisation of nutrients by the feto- placental tissues especially in sheep and rodents. 

The studies used were plenty.

In order to improve the robustness of the conclusions, authors should add the specific parts of search strategy, keywords, number of studies found and included/excluded and so on; secondly, authors should add the limitations of the study and; last, the final conclusions should be based on the type of studies they used. I full acknowledge, on the other hand, that this is a narrative review. 

Author Response

We thank the reviewer for their helpful comments. All changes to teh text have been highlighted in yellow

This review was not designed as a systematic review of the literature. Rather it is a narrative synopsis of in vivo studies in experimental animals that primarily use quantitative measurements to assess the effects of glucocorticoids (GCs)  on feto-placental metabolism impossible in pregnant women. Where human analogies are possible using perfused placenta or fresh placental explants or cell preparations in vitro, human data has been references when relevant.  This has now been made clear in the abstract and introduction (line 22-23 and 42-245). As indicated by the length of the reference list, the literature search was extensive and centered on the key words, and combinations thereof, listed on the front piece of the paper (together with rat, mouse and sheep).  The conclusions reiterate that the studies presented are primarily on experimental animals in vivo and indicate the specific species involved in the conclusion statement where relevant (lines 632-669).

All non-scientific spellings in the text have now been Americanized.

Reviewer 2 Report

i have read this paper with great interest, and highly value the paper as submitted. I only have two 'lines' of comments

firstly, can the authors add something on the mechanisms involved like changes in placental transporters for the prenatal part, or ? epigenetic changes to explain the long term effects ? 

second, this review has its focus on rodents and sheep, but 'sublimbic', the human perinatal setting remains present. I understood that especially sheep is more 'sensitive' to these effects. Could you comment on this, and how to convert this to the human setting ? do we need more data on long term outcome in CS exposed cases, with emphasis on indeed the 50 % inadverted exposed cases (cf recent WAPM guideline, DOI 10.1515/jpm-2022-0066, the FIGO good practice recommendation, DOI 10.1002/ijgo.13836.

Alternatively, this could be captured in the associated editorial, to put these findings into a human perspective, but that's an editorial decision. 

Author Response

We would like to thank the reviewer for their helpful comments, most of which have been incorporated into the revised text. All changes are highlighted in yellow.

We have added a a few sentences on the placental nutrient transporters in the molecular and cellular mechanisms section (lines 427-436). We have mentioned epigenetics again in the conclusion (lines 660-664) as well as in the original text (lines 415-421).  We feel that any further discussion of the epigenetic changes underlying long term developmental programming could be a review article in itself and is beyond the scope of the current manuscript. The human fetus is likely to be as sensitive to glucocorticoids (GCs)as the sheep fetus, if not more so given its haemochorial placenta. Certainly, antenatal treatment with synthetic GCs is unlikely to be as successful at reducing mortality in pre-term human neonates if cortisol was not involved in the normal prepartum metabolic adaptations in human infants. It is just that we know so much more about the effects of cortisol on the fetus when we can study it in utero in the conscious state as in chronically instrumental pregnant sheep.  We have added that more information is needed about the long term effects of intrauterine GC exposure in human infants to the final conclusions (line 660-664). See revised text below
